# Drug Solubility Correlation Using the Jouyban–Acree Model: Effects of Concentration Units and Error Criteria

**DOI:** 10.3390/molecules27061998

**Published:** 2022-03-20

**Authors:** Elaheh Rahimpour, Sima Alvani-Alamdari, William E. Acree, Abolghasem Jouyban

**Affiliations:** 1Pharmaceutical Analysis Research Center, Tabriz University of Medical Sciences, Tabriz 5165665811, Iran; rahimpour_e@yahoo.com; 2Infectious and Tropical Diseases Research Center, Tabriz University of Medical Sciences, Tabriz 5163639888, Iran; 3Department of Pharmaceutical Chemistry, Faculty of Pharmacy, Tabriz University of Medical Sciences, Tabriz 5166414766, Iran; sima_alvani@yahoo.com; 4Drug Applied Research Center, Tabriz University of Medical Sciences, Tabriz 5165665811, Iran; 5Department of Chemistry, University of North Texas, Denton, TX 76203, USA; bill.acree@unt.edu; 6Faculty of Pharmacy, Near East University, Nicosia P.O. Box 99138, North Cyprus, Mersin 10, Turkey

**Keywords:** solubility, prediction, correlation, unit expression, Jouyban–Acree model

## Abstract

An important factor affecting the model accuracy is the unit expression type for solute and solvent concentrations. One can report the solute and solvent concentration in various units and compare them with various error scales. In order to investigate the unit and error scale expression effects on the accuracy of the Jouyban–Acree model, in the current study, seventy-nine solubility data sets were collected randomly from the published articles and solute and solvent concentrations in the investigated systems were expressed in various units. Mass fraction, mole fraction, and volume fraction were the employed concentration units for the solvent compositions, and mole fraction, molar, and gram/liter were the investigated concentration units for the solutes. The solubility data, with various solute/solvent concentration units, were correlated using the Jouyban–Acree model, and the accuracy of each model for correlating the data was investigated by calculating different error scales and discussed.

## 1. Introduction

Solubility is an important physico-chemical property which is in demand for different applications in the pharmaceutical industries [1]. These applications are the proper solvent chosen for synthesis, extraction, purification, and dissolving media for assessing the biological activity of a drug/drug-like compound. The commonly used method to provide the solubility data of drugs in mono-solvents and binary and ternary mixtures is their experimental measurements, which is a costly and time-consuming procedure. Another main limitation of experimental measurements arises in early drug discovery studies. Only small amounts of the drug powder are available, and many experimental determinations need to be performed. As a possible solution, a number of mathematical models were reported for the correlation/prediction of the solubility data in mono-solvent or binary mixtures. These models for estimating the solubility of drugs were reviewed by various research groups [2,3,4,5,6]. In the pharmaceutical applications of these models, their accuracy, simplicity, and amount of required input data are important parameters in their acceptance by the pharmaceutical investigators. An important factor affecting the model accuracy is unit expression type for solute and solvent concentrations. One can report the solute and solvent concentration in various units (e.g., mass fraction, mole fraction, and volume fraction for solvents and mole fraction, molar, and gram/liter for solutes). In the case of solute concentration, molarity and gram/liter are volumetric scales related to the moles of the solute. In contrast, the mole fraction is a gravimetric scale and is related to the number of moles of both solute and solvents. In the case of solvent concentration, mass fraction and mole fraction are gravimetric scales, whereas volume fraction is a volumetric scale [7]. Gravimetric scales are relatively robust scales; however, the volumetric scale can be affected by temperature, due to the possible expansion of the solution, especially at higher temperatures. Different cosolvency models were used for correlating the solubility of solutes in solvent mixtures [8,9]. The Jouyban–Acree model was one of the most accurate models that has recently attracted more attention. In order to investigate the unit expression effects on the accuracy of the Jouyban–Acree model, the aims of this work were: (1) to collect solubility data for several drugs in different solvent mixtures and express them in various units; (2) to fit each data set to the Jouyban–Acree model with various units and compute the deviation of back-calculated data; and (3) to compare the suitability of various accuracy criteria.

## 2. Experimental Data Sets and Computational Methods

The collected solubility data sets from the literature (a total of 79 data sets) were fitted to the Jouyban–Acree model and explained in detail for each analysis. Drug concentrations were converted using the molecular masses of the drugs and/or the density values of the saturated solutions. The solvent compositions were converted employing the density of the solvent mixtures taken from the literature. Various combinations of the solute/solvent concentration units were analyzed in this work. Table 1 lists the details of these expressions. Obviously, all fraction concentration units, i.e., mole fraction, mass fraction, and volume fraction, varied in the range of 0.0–1.0. The minimum molar concentration of the investigated data points was 3.8 × 10^−6^ mole/L or 1.0 × 10^−3^ g/L (for sulfadiazine datum dissolved in water at 293.2 K, SN = 59) and the maximum value (16.2 mole/L correspond to 2786.6 g/L) belonged to sulfanilamide in 1,4-dioxane + water (SN = 70, 0.5 + 0.5 mole fractions) at 323.2 K. A major part of these wide variations was compensated in the two first terms of the Jouyban–Acree model, in the logarithmic scale, and the range of the variation of the obtained excess values were much narrower.

The Jouyban–Acree model, the most accurate cosolvency model [8], is described as follows:(1)$$\ln x_{m,T}=w_{1}.\ln x_{1,T}+w_{2}.\ln x_{2,T}+\frac{w_{1}.w_{2}}{T}\overset{np}{\underset{i=0}{\sum }}J_{i}.{\left(w_{1}-w_{2}\right)}^{i}$$
where *x*_1,*T*_, *x*_2,*T*_, and *x_m_*_,*T*_ represent the solubility of the solute in mono-solvents, one and two, and mixed solvents in various concentration units (in this work, mole fraction, molar, and g/L) at a temperature of ‘*T*’, respectively. The *w*_1_ and *w*_2_ stand for the concentrations of the mono-solvents, one and two, in the absence of the solute. In this work, these parameters are expressed in various units (mole fraction, mass fraction, and volume fraction). Terms of *J_i_* are the parameters of the model and are computed by regressing analysis of ($\ln x_{m,T}-w_{1}.\ln x_{1,T}-w_{2}.\ln x_{2,T}$) against ($\frac{w_{1}.w_{2}}{T}$), ($\frac{w_{1}.w_{2}\left(w_{1}-w_{2}\right)\;}{T}$), and ($\frac{w_{1}.w_{2}{\left(w_{1}-w_{2}\right)}^{2}\;}{T}$). The number of parameters (np) is usually two but, for some cases, up to three or even four can be used.

The experimental solubility data ($x_{}^{Exp.}$), in the current work, were fitted to the model and the back-calculated solubility data ($x_{}^{Cal.}$) were used to compute some indices of error evaluation, including the percentage of mean relative deviation (*MRD%*), relative mean square deviation of arithmetic scale (*RMSD*_1_), *RMSD* of logarithmic scale (*RMSD*_2_), error in arithmetic scale (*E*_1_), and error in logarithmic scale (*E*_2_), computed using Equations (2)–(6):(2)$$\%MRD=\frac{100}{N}\sum \left(\frac{\left|x_{}^{Cal.}-x_{}^{Exp.}\right|}{x_{}^{Exp.}}\right)$$
(3)$$RMSD_{1}=\sqrt{\frac{\sum_{i=1}^{N}{\left(x_{}^{Cal.}-x_{}^{Exp.}\right)}^{2}}{N}}$$
(4)$$RMSD_{2}=\sqrt{\frac{\sum_{i=1}^{N}{\left(lnx_{}^{Cal.}-lnx_{}^{Exp.}\right)}^{2}}{N}}$$
(5)$$E_{1}=\frac{\sum_{i=1}^{N}{\left|x_{}^{Cal.}-x_{}^{Exp.}\right|}^{}}{N}$$
(6)$$E_{2}=\frac{\sum_{i=1}^{N}{\left|lnx_{}^{Cal.}-lnx_{}^{Exp.}\right|}^{}}{N}$$
where *N* is the number of data points in each set.

## 3. Results and Discussion

The solubility data of each drug expressed in the units of mole fraction, molar, and gram/liter in the binary solvent mixtures with the solvent compositions expressed in various units of mole fraction, mass fraction, and volume fraction, defined as codes 1–9 (see Table 1 for details), were fitted to Equation (1). More details of the collected data sets are listed in Appendix A. The back-calculated data were used to compute various error evaluation criteria. When overall *MRD%* values (for codes 1–9) were classified according to the drug, the largest value was obtained for the ketoconazole data sets (overall *MRD%* = 25.7) and the smallest value was observed for the dapsone data sets (overall *MRD%* = 4.9). The obtained error values for each numerical method, which were expressed in *MRD%*, are listed in Appendix A. The largest *MRDs%* for codes 1–3 were observed for the solubility of ketoconazole in the carbitol + water system (SN = 12) and those for codes 4–9 were obtained for ketoconazole in the acetonitrile + water system (SN = 11).

Figure 1 illustrates the overall *MRD%* and their standard deviations (SDs) for 79 data sets and different numerical analysis codes. As can be seen from the results, there was no significant difference in overall *MRD%* values for codes 1–3 and 4–9; however, there was a significant difference among these subgroups. These results mean that the drug concentration was not an affecting parameter on the fitness capability of the Jouyban–Acree model when *MRD%* was considered as an error criterion; however, the concentration of the solvents in the absence of the drug might affect the fitness of the model to the experimental data. Careful examination of the distributions of the various solvent compositions revealed that the mean value of the mole fractions was 0.36, whereas those of the mass and volume fractions were 0.52 and 0.52. Our earlier observations showed that, with the equal distances among the fractions (i.e., mean fraction of 0.50), the Jouyban–Acree model provided the most accurate correlations, and the observed differences among codes 1–3 (expressed as mole fraction) with codes 4–9 (expressed in volume or mass fractions) could be justified by the skewness of the mole fractions. Another difference in these analyses was several variations in the numerical values of the model constants of Equation (1) and, also, the number of significant *J* terms of the Jouyban–Acree model. As an example, the *J*_0_, *J*_1_, *J*_2_, and the obtained *MRD%* values for the solubility data of sulfadiazine in acetonitrile + methanol mixtures (SN = 60) are listed in Table 2. The mean of mole (0.46), mass (0.50), and volume (0.50) fractions of the solvent composition in this set was relatively equal. Similar investigations were carried out on the solubility data of paracetamol in PEG 400 + water (SN = 55), with the mean of the mole (0.14), mass (0.50), and volume (0.48) fractions of the solvent composition. The highest deviations from 0.50 was observed for the mole fraction data, and the obtained *MRD%* for code 1 was 14.2%. Meanwhile, the corresponding values for mass (code = 4) and volume (code = 7) fractions were 3.3 and 3.0%. In another data set, i.e., meloxicam in ethanol + ethyl acetate (SN = 55), with the mean of the mole (0.50), mass (0.58), and volume (0.60) fractions of the solvent composition, the *MRDs%* for codes 1, 4, and 7 were 14.2, 3.3, and 3.0%, respectively.

Table 3 lists the effects of different numbers of the *J* terms and the *MRD%* values for SN = 60. As was expected, employing more curve-fitting parameters, i.e., the *J* terms, more accurate correlations could be obtained. According to the theoretical justification of the Jouyban–Acree model [8,9], the *J* terms represent the non-ideal mixing behavior of the solution. For ideal mixing behavior, all *J* terms were non-significant constants and the Jouyban–Acree model reduces to the Yalkowsky model [10]. The Yalkowsky model is an algebraic linear model which consider an ideal mixing for solvent mixtures without any energy exchanges. This model is expressed as:(7)$$\ln x_{m}=w_{1}.\ln x_{1}+w_{2}.\ln x_{2}$$

Appendix A lists the obtained results, employing *RMSD*_1_ as an accuracy criterion. The overall 10^5^ *RMSD*_1_ values varied from 125.3 (for code 7) to 8,948,236 (for code 3). Concerning the solvent compositions, the order of *RMSD*_1_ values for the mole fraction solubility of the drugs was volume fraction (125.3) < mass fraction (132.8) < mole fraction (312.0). The corresponding orders concerning the molar and g/L drugs’ concentrations were volume fraction (3302.3) < mass fraction (3354.0) < mole fraction (8948.2) and volume fraction (3,302,273) < mass fraction (3,353,971) < mole fraction (8,948,236). It seems that the numerical values of drug solubility in the saturated solutions were the governing parameters in *RMSD*_1_ calculations, in which the overall 10^5^ *RMSD*_1_ for the drugs’ mole fraction solubilities was 190.0, or $\frac{125.3+132.8+312.0}{3}$, and those of molar and g/L were 5201.5 and 5,201,493, respectively. The largest 10^5^ *RMSD*_1_ values (for codes 1–3, 5, 6, 8, and 9) were observed for the solubility of sulfanilamide in the 1,4-dioxane + water system (SN = 70) and was obtained for sulfadiazine in 1-propanol + water (SN = 59) for codes 4 and 7.

Appendix A reports the *RMSD*_2_ accuracy criterion for the investigated systems. The overall 100 *RMSD*_2_ values varied from 10.5 (for codes 8 and 9) to 19.8 (for code 1). Concerning the solvent compositions, the order of 100 *RMSD*_2_ values for the mole fraction solubility of the drugs was volume fraction (10.5) < mass fraction (10.9) < mole fraction (19.8). The corresponding orders concerning the molar and g/L drugs’ concentrations were volume fraction (10.5) < mass fraction (11.0) < mole fraction (19.4) and volume fraction (10.5) < mass fraction (11.1) < mole fraction (19.4). Similar to the *RMSD*_1_ case, the numerical values of drug solubility in the saturated solutions were the governing parameters in *RMSD*_2_ calculations in which the overall 100 *RMSD*_2_ were 13.7, 13.6, and 13.7, respectively, for the drugs’ mole fraction, molar, and g/L solubilities. The largest 100 *RMSD*_2_ value (for codes 1–3) was observed for the solubility of ketoconazole in carbitol + water system (SN = 12) and was obtained for ketoconazole in NMP + ethanol (SN = 14) for codes 4–9.

Appendix A lists the details of *E*_1_ for different codes where the largest *E*_1_ values were observed for the solubility of sulfanilamide in 1,4-dioxane + water (SN = 70) for codes 1–9. Table 4 lists the overall *E*_1_ values obtained for the various drugs investigated according to the investigated codes. Appendix A reports the details of the *E*_2_ values.

*E*_1_ and *E*_2_ are the absolute error, or variances in the arithmetic and logarithmic scales, respectively. The absolute error uses the same scale as the data being measured. Therefore, in the case of solubility with the g/L unit (especially in the arithmetic scale), the high values can be recorded as error criteria that make the data comparison difficult. *RMSD*_1_ and *RMSD*_2_ are root-mean-square deviations in the arithmetic and logarithmic scales, respectively. These error criteria are the mean square root of the variance and, similar to absolute error, are related to units of measurements. However, *MRD%,* as a mean relative deviation, facilitates the comparison between datasets or models with different scales due to normalizing the data by dividing the variance to the observed values. From this point of view, the *MRD%* definition is similar to %*RSD* (relative standard deviation which is used as a repeatability and reproducibility index for repeated measurements) and may be the best error criterion.

Furthermore, as the solubility data for the investigated systems (different solutes and different solvents) lies in different data value ranges and with considering the magnitude of data for high soluble compounds which show a high absolute error for both arithmetic and logarithmic scales, the comparison between different systems for finding the system with high error is not possible. Herein, *MRD%* can be a helpful error metric for the comparison of different systems. This is because this metric, with its normalizing data property, puts the data in a similar and comparable range.

Correlations between various error criteria against *MRD%* are shown in Figure 2. Code 4 values were chosen as the reprehensive one for showing the correlations. As presented in Figure 2, good correlations were observed between the *E*_2_ and *RMSD*_2_ error criteria vs. *MRD%,* so that the data scattered around the line. However, in the case of *RMSD*_1_ and *E*_1,_ some significant deviations were observed when the models assessed using the *MRD%* criterion.

In another effort, the effect of outlier data points on the error indices’ behavior were also investigated. For this purpose, we intentionally changed a datum in several reported data sets and studied the trend of each error metric. Code 4 values were, again, chosen as the reprehensive one for showing the correlation. For example, the solubility data value for dapsone in the mixture of ethanol + water (SN = 5) at 298.2 K in the ethanol mass fraction of 0.5 (i.e., 0.000930) was changed to 0.00930. The error increase was from 8.9% to 10.5 for *MRD%*, from 24.74 to 83.28 for 10^5^ *RMSD*_1_, from 1.37 to 2.24 for *E*_1_, from 11.52 to 24.46 for 100 *RMSD*_2,_ and from 0.089 to 0.11 for *E*_2_. In another investigation, the solubility data value for naproxen in the mixture of ethylene glycol and ethanol at 298.2 K in an ethylene glycol mass fraction of 0.5 (i.e., 0.0135) was changed to 1.35. The error increase was from 1.7% to 15.46 for *MRD%*, from 27.36 to 17959.59 for 10^5^ *RMSD*_1_, from 1.97 to 260.33 for *E*_1_, from 2.4 to 60.30 for 100 *RMSD*_2,_ and from 0.02 to 0.2 for *E*_2_. Large deviations were observed for overestimated/underestimated data points, but all error criteria could be employed to detect outliers.

## 4. Conclusions

In the current study, the Jouyban–Acree model was used to correlate some solubility data sets at various binary solvent mixtures with different solute/solvent concentration units and to compare the suitability of various units with computing several accuracy criteria. The obtained results show that *MRD*% can be the best error metric that facilitates the comparison between datasets, or models, with different scales due to normalizing the data. Considering this error criterion for back-calculation data with various solute/solvent concentrations, the results show that concentration units cannot affect the fitness capability of the Jouyban–Acree model. Meanwhile, the concentration of the solvents in the absence of the drug might affect the fitness of the model to the experimental data. The number of curve-fit parameters of the Jouyban–Acree model was also affected by solute/solvent concentration expressions. However, this incompatibility can be compensated with the definition of the equal distances among the fractions for each selected solvent composition unit.

## Figures and Tables

**Figure 1 molecules-27-01998-f001:**
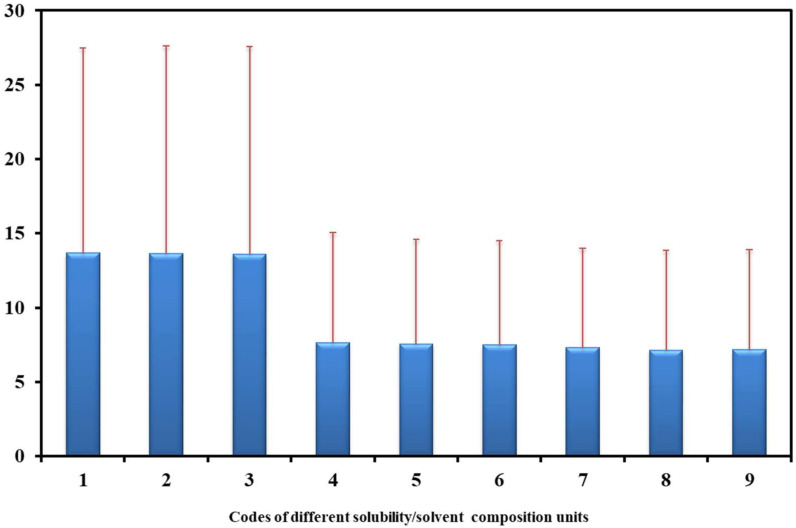
Overall *MRD%* and their standard deviations (SDs) using Equation (1) for investigated data sets.

**Figure 2 molecules-27-01998-f002:**
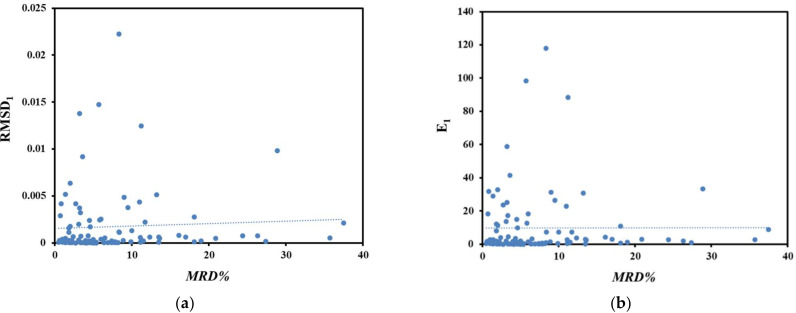
Correlations between various error criteria with the *MRD%* values for back-calculated data with Equation (1); (**a**) *RMSD*_1_ vs. *MRD%*, (**b**) *E*_1_ vs. *MRD%*, (**c**) *RMSD*_2_ vs. *MRD%* and (**d**) *E*_2_ vs. *MRD%*.

**Table 1 molecules-27-01998-t001:** The codes of different solubility/solvent composition units.

Drug Concentration → Solvent Composition ↓	Mole Fraction	Molar	Gram/Liter
Mole fraction	1	2	3
Mass fraction	4	5	6
Volume fraction	7	8	9

**Table 2 molecules-27-01998-t002:** Model constants for the solubility data of sulfadiazine in acetonitrile + methanol mixtures expressed in different concentration units and the obtained mean relative deviations.

Code↓/Constants→	*J* _0_	*J* _1_	*J* _2_	*MRD%*
1	897.693	−1417.112	1671.752	7.5
2	888.072	−1416.471	1675.101	7.5
3	888.650	−1417.871	1673.184	7.5
4	967.341	−1304.172	1264.565	5.4
5	976.557	−1303.853	1267.542	5.3
6	977.118	−1304.945	1265.856	5.3
7	969.304	−1300.003	1252.338	5.3
8	979.098	−1299.620	1255.305	5.3
9	979.657	−1300.703	1253.629	5.3

**Table 3 molecules-27-01998-t003:** Effects of different number of the *J* terms on the fitness of solubility data of sulfadiazine in acetonitrile + methanol mixtures expressed in different concentration units and the obtained mean relative deviations.

Code↓/Constants→	*J* _0_	*J* _1_	*J* _2_	*J* _3_	*J* _4_	*MRD%*
1	934.547	−937.019	1130.686	−1443.763	848.997	4.8
2	924.773	−973.122	1136.587	−1441.527	844.546	4.8
3	925.569	−936.743	1131.213	−1446.887	850.349	4.8
4	967.341	−1030.624	1264.565	−821.069	0 ^a^	4.4
5	976.557	−1031.718	1267.542	−816.830	0 ^a^	4.4
6	977.118	−1030.517	1265.856	−823.711	0 ^a^	4.4
7	969.117	−1032.713	1253.652	−802.289	0 ^a^	4.4
8	978.912	−1033.755	1256.612	−798.012	0 ^a^	4.3
9	979.469	−1032.538	1254.947	−804.914	0 ^a^	4.3

^a^ Not significant (*p* > 0.05).

**Table 4 molecules-27-01998-t004:** *E*_1_ values for different drugs according to numerical analyses codes 1–9.

					Code				
Drug	1	2	3	4	5	6	7	8	9
Dapsone	18.1	460.6	114,476.4	3.6	92.1	22,959.0	2.8	71.1	17,776.6
Ketoconazole	6.4	141.0	74,944.3	2.9	71.5	37,812.4	2.7	65.2	33,861.6
Meloxicam	0.5	7.3	2346.6	0.3	4.9	1721.5	0.3	4.5	1578.1
Mesalazine	1.2	26.2	4014.0	0.7	16.6	2538.5	0.8	18.2	2327.3
Naproxen	6.1	138.9	31,961.8	4.7	102.0	23,385.4	4.2	92.5	21,213.3
Paracetamol	38.9	928.3	140,713.0	26.4	684.5	74,475.9	26.5	708.4	107,100.9
Sulfadiazine	2.0	43.2	10,817.6	3.1	54.0	13,521.5	3.1	53.7	13,455.9
Sulfanilamide	77.7	2658.7	457,833.6	22.4	666.0	114,687.7	22.0	649.5	111,850.9

## Data Availability

On request from the corresponding author.

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
