# Peer review of "Drug Solubility Correlation Using the Jouyban–Acree Model: Effects of Concentration Units and Error Criteria"

_molecules, 2022, doi:10.3390/molecules27061998_

Round 1

Reviewer 1 Report

In the present paper authors discussed applicability of different error scales to evaluate the accuracy of solubility correlation according to Jouyban-Acree Model in binary solvent mixtures. Authors highlighted the efficiency of mean relative deviation for comparison of between datasets or models with different scales.

Before publication, I believe the paper needs some changes.

Abstract contains the description of what has been done, but the problem and purpose are unclear.

Some sentences need to be re-written to make the manuscript more reader-friendly and the language less formal, p. 18-21, p. 77-79, 197-199.

Why is code 4 more reprehensive than others?

There is inconsistency in verb tenses within a sentence, e.g.:

“In the current study seventy-nine solubility data sets were collected randomly from the published articles and solute and solvent concentrations in the investigated systems are expressed in various units.”

100000 should be better represented as 10 powered by 5 (e.g. 100000 RMSD, line 214).

Line 52, replace ; with :

Author Response

Dear Prof. Nikolic 

Many thanks for your kind attention and the comments of the reviewers. We revised the manuscript molecules-1633277   concerning the comments. We wish to thank the referees for their time in forming a constructive critique of our work. We value the comments and suggestions made by referees and we address each of their points in sequence below. All modifications were written in RED font on the revised manuscript.

Looking forward to hearing from you.

Sincerely yours

  1. Jouyban

The corresponding author

Reviewers' comments:

Reviewer #1: In the present paper authors discussed applicability of different error scales to evaluate the accuracy of solubility correlation according to Jouyban-Acree Model in binary solvent mixtures. Authors highlighted the efficiency of mean relative deviation for comparison of between datasets or models with different scales.

Before publication, I believe the paper needs some changes.

Abstract contains the description of what has been done, but the problem and purpose are unclear.

Thanks for valuable comment. Some explanation about problem and purpose is added. “An important factor affecting the model accuracy is unit expression type for solute and solvent concentrations. One can report the solute and solvent concentration in various units and compare them with various error scales. In order to investigate the unit and error scale expression effects on the accuracy of the Jouyban-Acree model, in the current study…..”

Some sentences need to be re-written to make the manuscript more reader-friendly and the language less formal, p. 18-21, p. 77-79, 197-199.

Thanks for careful review. The sentences are modified.

Why is code 4 more reprehensive than others?

The solute and solvent units related to code 4 are units that are mostly used in the solubility articles. So, we used this code as a reprehensive code. 

There is inconsistency in verb tenses within a sentence, e.g.:

 “In the current study seventy-nine solubility data sets were collected randomly from the published articles and solute and solvent concentrations in the investigated systems are expressed in various units.”

The manuscript thoroughly checks and some corrections are done that marked with red font. 

100000 should be better represented as 10 powered by 5 (e.g. 100000 RMSD, line 214).

Ok. It is corrected.

Line 52, replace ; with :

It is corrected. “….to collect solubility data for several drugs in different solvent mixture and express them in various units….”

Reviewer 2 Report

I think the analysis you made in this article will be very useful for the authors that will use the Jouyban-Acree model in their studies. 

I have only an observation:

In lines 145-146 you explained that "For ideal mixing behavior, all J terms are non-significant constants and the Jouyban-Acree model reduces to the algebraic mixing model of Yalkowsky." You should explain or you should give us a reference for what the Yalkowsky model is. 

So, I think that the paper can be considered for publication after a minor revision.

Author Response

Dear Prof. Nikolic 

Many thanks for your kind attention and the comments of the reviewers. We revised the manuscript molecules-1633277   concerning the comments. We wish to thank the referees for their time in forming a constructive critique of our work. We value the comments and suggestions made by referees and we address each of their points in sequence below. All modifications were written in RED font on the revised manuscript.

Looking forward to hearing from you.

Sincerely yours

  1. Jouyban

The corresponding author

Reviewers' comments:

Reviewer #2: I think the analysis you made in this article will be very useful for the authors that will use the Jouyban-Acree model in their studies.

I have only an observation:

In lines 145-146 you explained that "For ideal mixing behavior, all J terms are non-significant constants and the Jouyban-Acree model reduces to the algebraic mixing model of Yalkowsky." You should explain or you should give us a reference for what the Yalkowsky model is.

Thanks for careful review. Some explanation along with original form of equation and related reference are added to the manuscript.

“….all J terms are non-significant constants and the Jouyban-Acree model reduces to the Yalkowsky model [10]. The Yalkowsky model is an algebraic linear model which consider an ideal mixing for solvent mixtures without any energy exchanges. This model is expressed as:

                                                                                   (7)”

So, I think that the paper can be considered for publication after a minor revision.